# Comparative Compositions of Grain of Bread Wheat, Emmer and Spelt Grown with Different Levels of Nitrogen Fertilisation

**DOI:** 10.3390/foods12040843

**Published:** 2023-02-16

**Authors:** Alison Lovegrove, Jack Dunn, Till K. Pellny, Jessica Hood, Amanda J. Burridge, Antoine H. P. America, Luud Gilissen, Ruud Timmer, Zsuzsan A. M. Proos-Huijsmans, Jan Philip van Straaten, Daisy Jonkers, Jane L. Ward, Fred Brouns, Peter R. Shewry

**Affiliations:** 1Rothamsted Research, Harpenden AL5 2JQ, UK; 2Life Sciences, University of Bristol, 24 Tyndall Avenue, Bristol BS8 1TQ, UK; 3BU Bioscience, Plant Sciences Group, Wageningen University and Research, 6700 AA Wageningen, The Netherlands; 4Plant Breeding, Wageningen University & Research, 6700 AJ Wageningen, The Netherlands; 5Business Unit Field Crops, Wageningen University & Research, 8200 AK Lelystad, The Netherlands; 6Nederlands Bakkerij Centrum, Agro Business Park 75-83, 6708 PV Wageningen, The Netherlands; 7Division Gastroenterology-Hepatology, Department of Internal Medicine and School for Nutrition and Translational Research in Metabolism (NUTRIM), Maastricht University, 6200 MD Maastricht, The Netherlands; 8Department of Human Biology, Faculty of Health, Medicine and Life Sciences, School for Nutrition and Translational Research in Metabolism (NUTRIM), Maastricht University, 6700 MD Maastricht, The Netherlands

**Keywords:** bread wheat, emmer, spelt, fibre, metabolites, minerals, phenolics, fertilisation, health benefits

## Abstract

Five cultivars of bread wheat and spelt and three of emmer were grown in replicate randomised field trials on two sites for two years with 100 and 200 kg nitrogen fertiliser per hectare, reflecting low input and intensive farming systems. Wholemeal flours were analysed for components that are suggested to contribute to a healthy diet. The ranges of all components overlapped between the three cereal types, reflecting the effects of both genotype and environment. Nevertheless, statistically significant differences in the contents of some components were observed. Notably, emmer and spelt had higher contents of protein, iron, zinc, magnesium, choline and glycine betaine, but also of asparagine (the precursor of acrylamide) and raffinose. By contrast, bread wheat had higher contents of the two major types of fibre, arabinoxylan (AX) and β-glucan, than emmer and a higher AX content than spelt. Although such differences in composition may be suggested to result in effects on metabolic parameters and health when studied in isolation, the final effects will depend on the quantity consumed and the composition of the overall diet.

## 1. Introduction

Wheat is the major staple crop in temperate countries, with annual global yields exceeding 700 million tonnes. About 95% of the total production is hexaploid bread wheat (*Triticum aestivum* L. subsp. *aestivum*, genome constitution AABBDD) which originated about 10,000 years ago, with most of the remaining 5% production being tetraploid pasta wheat (*Triticum turgidum* L. subsp. *durum*) (AABB genomes). Bread and durum wheats are “free threshing” (the hulls being readily separated from the grain during harvest), which is regarded as an advanced trait. However, both bread and pasta wheats have been subjected to intensive breeding, focusing on improving their agronomic performance and increasing their yield and quality (for making bread and pasta, respectively). Hence, although there is wide genetic variation in both species, modern cultivars (those developed by scientific breeding during the last few decades) tend to be less genetically diverse than older cultivars and traditional types of wheat dating from before the application of breeding (called land races) [1]. 

Ancient wheats are diploid einkorn (*T. monococcum*, L., AA genome), tetraploid emmer (*T. turgidum* L. subsp. *dicoccum* Thell., AABB genomes) and hexaploid spelt (*T. aestivum* L. subsp. *spelta* Thell, AABBDD genomes) and are generally hulled as opposed to free threshing. Although the term “ancient” is taken to imply that the genotypes grown today are similar to those grown in antiquity, this is certainly not the case, at least for spelt and emmer. Einkorn is a distinct species, which includes cultivated and wild forms, while emmer and spelt are subspecies of *T. turgidum* and *T. aestivum*, respectively. Furthermore, modern commercial cultivars of spelt may have introgressions (transfer of genetic information) from bread wheat due to cross-breeding, while all types of wheat cultivated today have been grown and hence selected (either unconsciously or deliberately) over thousands of years [2].

The composition of wheat grain is determined by the genotype, the environment, the farming system and the interactions between the genotype and these factors. Environmental factors are particularly important when comparing ancient and modern wheats, as the former are often grown in organic or low input systems with low nitrogen application to avoid lodging (bending of the plant at or near ground level, making harvest difficult and often leading to premature germination of grains) while modern semi-dwarf wheats are more usually grown in intensive high input systems with high nitrogen fertilisation [3].

The breeding and selection of modern bread wheats have focused on increasing yield and improving breadmaking quality, which is largely determined by the content and composition of gluten proteins. It has therefore been suggested that this has led to modern wheats having lower contents of micronutrients (minerals, vitamins) and bioactive components (phytochemicals) and higher contents of proteins that may lead to adverse reactions and diseases such as coeliac disease, wheat allergy and non-coeliac wheat sensitivity [4,5]. Hence, there has been increased interest in ancient wheats, which are assumed to have more favourable compositions for health [2].

We have therefore carried out detailed analyses of grain samples of three commercial cultivars of emmer and five cultivars each of spelt and bread wheat. All of the cultivars are adapted to Northern Europe and corresponded, with one substitution due to unavailability, to those selected to compare the effects of breads on health as part of the “Well-on-Wheat?” research consortium programme (https://www.wellonwheat.org, accessed on 1 February 2023). These samples were grown in replicate field trials in two Northern European countries (the UK and the Netherlands) for two years with low (100 kg/Ha) and high (200 kg/ha) applications of nitrogen fertilisation to reflect the different inputs used for ancient and modern wheats. Wholemeal samples were analysed for the major types of dietary fibre in white flour (arabinoxylan and β-glucan), polar metabolites, protein (as nitrogen), phenolics and mineral micronutrients to identify differences in composition. 

## 2. Materials and Methods

### 2.1. Grain Samples

Commercial samples of five cultivars each of bread wheat (RAGT Reform, Capo, Bernstein, Kometus, Akteur) and spelt (Comburger, Zollernspelz, Attergauer, Bauländer Spelz, Franckenkorn) and three cultivars of emmer (Ramses, Roter Heidfelder, Späths Albjuwel) were grown in two years (2017–2018 and 2018–2019) at two nitrogen levels (100 and 200 kg N/Ha) in Flevoland (WUR Field crops, Lelystad, Flevoland, Netherlands, 52°53′94.69″ N, 5°56′56.77″ E) in three randomised replicate 6 × 1.5 m plots and at Rothamsted Research (Harpenden, Hertfordshire, UK, 51°48′19.79″ N 0°21′11.39″ W) in three randomised replicate plots of 1 × 1 m. All trials were autumn sown, but sowing, fertiliser application and harvest dates varied between sites and years, according to local conditions. Standard agronomic practices for the two sites were used. Grain samples of emmer and spelt were mechanically dehulled. Whole grain samples at about 14% water content were milled in two stages: firstly, a Retsch ZM 200 Model Ultra-Centrifugal Mill (Retsch Gmbh, Dusselgorf, Germany) using a 0.5 mm ring sieve and then a Glen Creston Ball Mill Retsch Gmbh, Dusseldorf, Germany) using 5 ball bearings in a 5 cm diameter canister for 4 min for each sample. 

### 2.2. Genotyping 

The samples were genotyped using the Axiom 35k Wheat Breeders Genotyping Array (Thermo Fisher Scientific, Inc., Waltham, MA, USA) using the Affymetrix GeneTitan (Thermo Fisher Scientific, Inc.) [6]. Alleles were identified using the Affymetrix proprietary software package Axiom Analysis Suite V4.0.3.3 (Thermo Fisher Scientific, Inc.) and prior model ‘Axiom_WhtBrd-1.r3’. A Dish QC threshold of 0.8 and call rate cut-off of 90% were used to adjust for hybridisation rates of spelt and emmer to the array. A distance matrix was generated from the scores using R package SNPRelate (Bioconductor Open Source, Harvard, MA, USA) [7]. The first two Principal Components accounting for over 25% of the variance (PC1:19.76%; PC2:5.53%) were plotted as a PCA plot.

### 2.3. Enzyme Fingerprinting of Arabinoxylan and β-Glucan

Three technical replicates of flour were digested with endoxylanase and lichenase (β-glucanase) to release arabinoxylan oligosaccharides (AXOS) from arabinoxylan (AX) and gluco-oligosaccharides (GOS) from β-glucan, respectively [8]. The oligosaccharides were separated using a 2 mm × 250 mm Carbopac PA-1 (Dionex) column [8] (dx.doi.org/10.17504/protocols.io.babriam6, accessed on 1 February 2023). The areas under the oligosaccharide peaks were combined to give total AX and total β-glucan (expressed in arbitrary units), respectively.

### 2.4. NMR Spectroscopy of Polar Metabolites

Sample preparation for ^1^H-NMR was carried out as described by [9]. Signal intensities for characteristic spectral regions for 29 major metabolites were compared with a library of spectra of standards analysed under the same conditions.

### 2.5. Mineral Analysis 

Nitrogen was determined on each biological replicate by Dumas combustion, using a Leco combustion analyzer (Leco Corp., St. Paul, MN, USA). Iron and zinc were determined by Optima 7300 DV Inductively Coupled Plasma–Optical Emission Spectrometer (ICP–OES) (Perkin Elmer, Waltham, MA, USA) after digestion with nitric and perchloric acids. Certified external standards and in-house standards were used to monitor performance using Shewhart Control Charts.

### 2.6. Total Phenolics

Total phenolics were determined based on [10]. Triplicate 75 mg samples of each biological replicate were vortexed with 1.5 mL acidified methanol and then mixed at 850 rpm on an Eppendorf Thermomixer (Eppendorf Ltd., Stevenage, UK) for 2 h at 23 °C. After centrifugation (Eppendorf Ltd., Stevenage, UK) at 5000× *g* for 10 min, 1 mL of supernatant was removed into a fresh Eppendorf tube and 200 µL aliquots mixed with 1.5 mL of ×10 diluted Folin–Ciocalteau reagent (Sigma-Aldrich, St. Louis, MO, USA) and left to stand for 5 min. 1.5 mL of 6% (*w*/*v*) aqueous sodium carbonate solution was added, mixed and stood at room temperature for 90 min. The absorbance at 725 nm was then measured (Jenway 6715 UV/Vis spectrophotometer, Cole-Parmer, St Neots, UK), and the concentration of phenolics was calculated using ferulic acid as a standard (Sigma-Aldrich) and a standard curve from 20, 40, 100, 150 and 200 µg/mL with three technical replicates of each point.

### 2.7. Statistical Analysis

All data were analysed using analysis of variance (ANOVA) in Genstat 21 (VSN International, Hemel Hempstead, UK). The block structure was Trial/Block/Subblock, where Trial captures the location and year of each trial. There are 3 blocks within each trial and 2 sub-blocks within each block to which the Nlevel treatment was applied. Lines were considered to be applied to plots within each sub-block. The treatment structure was Nlevel*(Grain/(cultivarBreadwheat + cultivarEmmer + cultivarSpelt)) where the factor Grain tests for differences between the three grain types. The nested factors cultivarBreadwheat, cultivarEmmer and cultivarSpelt test for differences between the lines within each grain type. The Nlevel factor tests for differences between the two Nlevels and their interactions with grain type and lines are also included. Some variables were transformed in order to meet the normality and homoscedasticity assumptions of the analysis. Means and 95% confidence intervals are given in Tables 1 and 3 while the transformations used are indicated in Appendix A of means and *p* values.

Principal component analysis (PCA) and orthogonal partial least squares-discrimination analysis (OPLS-DA) were carried out using SIMCA-P software (version 13, MKS Umetrics) (Sartorius UK Ltd., Epsom, UK).

## 3. Results

Five cultivars each of bread wheat (RAGT Reform, Capo, Bernstein, Kometus, Akteur) and spelt (Comburger, Zollernspelz, Attergauer, Bauländer Spelz, Franckenkorn) and three cultivars of emmer (Ramses, Roter Heidfelder, Späths Albjuwel) were selected, all of which have been grown commercially in Northern Europe. The genomic relationships between the cultivars were determined using the Axiom 35k Wheat Breeders Genotyping Array, comprising 35,143 single nucleotide polymorphism (SNP) markers. Principal component analysis (PCA) showed a clear separation of the three cereal types, confirming that the spelt lines used did not have recent introgressions from bread wheat (Figure 1). 

### 3.1. Grain Composition 

In order to determine the variation in composition within and between bread wheat, spelt and emmer, the four environments (site × year combinations) were treated as blocks in the statistical analyses. In addition, to compare the effects of nitrogen fertilisation on grain composition, the data for the 100 kg/Ha and 200 kg/Ha treatments were analysed separately and compared formally by including nitrogen as a factor in the statistical analyses. The ranges of contents are illustrated in Figures 2, 3 and 5, while Table 1, Table 2, Table 3 and Table 4 and Appendix A present means, 95% confidence intervals, SEMs and observed statistical significance determined by ANOVA.

The groups of components are discussed below.

### 3.2. Protein and Minerals

Grain protein content (determined as N × 5.7) (Figure 2A, Table 1) overlapped in range between bread wheat, spelt and emmer but was highest in spelt and lowest in bread wheat. It was also higher at 200 kg/Ha than at 100 kg/Ha. ANOVA (Table 2) showed statistically significant differences in protein content between bread wheat, emmer and spelt and statistically significant effects of nitrogen on their protein contents. By contrast, the three cereal types did not differ significantly in their response to nitrogen, and there was little variation in the effects of nitrogen between the cultivars within a single cereal type. 

**Figure 2 foods-12-00843-f002:**
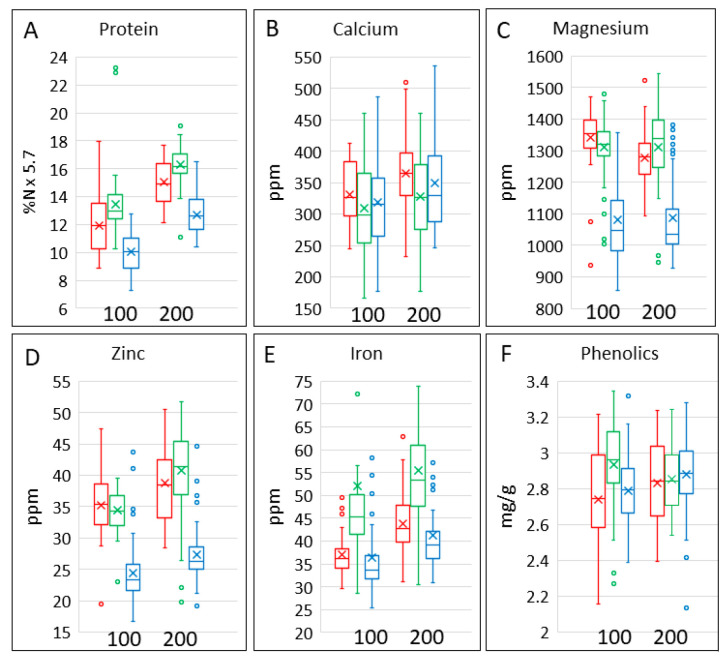
Contents of protein, minerals and total phenolics in grains of the three cereal types grown in four environments. Colour code: red, emmer; green, spelt; blue, bread wheat. The bar shows the range of the whole dataset. The box shows the middle two quartiles, separated by the horizontal line, which is the median, and the vertical lines are the upper and lower quartiles, respectively. Outliers are shown as circles. The x is the mean average. All analyses are expressed on a dry weight basis.

**Table 1 foods-12-00843-t001:** Means and 95% confidence intervals (in parentheses) of contents of selected minerals, metabolites and groups of metabolites in grain of the three types of wheat grown with 100 and 200 kg N/Ha.

	100 kg N/Ha			200 kg N/Ha		
	Bread Wheat	Emmer	Spelt	Bread Wheat	Emmer	Spelt
%N	1.764 (1.68, 1.847)	2.095 (2.001, 2.189)	2.354 (2.27, 2.437)	2.23 (2.146, 2.313)	2.636 (2.542, 2.73)	2.857 (2.773, 2.94)
Ca	311.6 (300.2, 323.5)	327.7 (312.3, 343.8)	301.8 (290.7, 313.3)	341.6 (329.1, 354.6)	358.5 (341.7, 376.1)	320.2 (308.5, 332.4)
Fe	35.33 (33.36, 37.42)	36.65 (34.01, 39.5)	47.65 (44.98, 50.46)	40.04 (37.8, 42.4)	43.22 (40.11, 46.58)	54.01 (51, 57.21)
Mg	1081 (1054, 1108)	1341 (1309, 1374)	1311 (1284, 1338)	1087 (1060, 1114)	1278 (1246, 1311)	1312 (1285, 1339)
Zn	24.41 (22.78, 26.04)	35.22 (33.34, 37.11)	34.37 (32.74, 36)	27.27 (25.64, 28.9)	38.76 (36.88, 40.65)	40.78 (39.15, 42.41)

total phenolics	2786 (2732, 2840)	2742 (2677, 2807)	2934 (2880, 2987)	2879 (2826, 2933)	2830 (2765, 2894)	2854 (2800, 2908)

raffinose	5.95 (5.819, 6.082)	7.077 (6.908, 7.245)	6.916 (6.785, 7.048)	5.944 (5.813, 6.076)	6.465 (6.296, 6.633)	6.743 (6.611, 6.874)
asparagine	0.4907 (0.4681, 0.5144)	0.6588 (0.6195, 0.7007)	0.6981 (0.666, 0.7318)	0.5664 (0.5403, 0.5937)	0.7803 (0.7337, 0.8298)	0.8648 (0.825, 0.9065)
glycine betaine	1.311 (1.262, 1.362)	1.49 (1.432, 1.548)	1.606 (1.551, 1.662)	1.307 (1.257, 1.357)	1.466 (1.41, 1.525)	1.537 (1.483, 1.591)
choline	0.1748 (0.1711, 0.1785)	0.2148 (0.2101, 0.2195)	0.2137 (0.21, 0.2174)	0.1854 (0.1817, 0.1891)	0.2207 (0.216, 0.2254)	0.2229 (0.2192, 0.2266)
galactinol	0.2224 (0.2122, 0.2328)	0.3976 (0.3795, 0.4161)	0.2692 (0.2579, 0.2806)	0.2223 (0.2121, 0.2327)	0.3657 (0.3484, 0.3834)	0.2751 (0.2637, 0.2867)
inositol	3.08 (3.024, 3.136)	3.216 (3.141, 3.292)	3.421 (3.365, 3.477)	3.097 (3.041, 3.153)	3.096 (3.021, 3.171)	3.25 (3.194, 3.306)

total amino acids	5.753 (5.618, 5.891)	6.522 (6.331, 6.719)	6.607 (6.452, 6.766)	6.084 (5.941, 6.23)	6.724 (6.528, 6.927)	6.951 (6.788, 7.118)
total organic acids	2.014 (1.934, 2.093)	2.247 (2.156, 2.337)	2.133 (2.053, 2.213)	2.048 (1.968, 2.128)	2.321 (2.231, 2.411)	2.245 (2.166, 2.325)
total methyl donors	1.482 (1.436, 1.529)	1.702 (1.644, 1.761)	1.814 (1.758, 1.871)	1.486 (1.44, 1.533)	1.685 (1.628, 1.744)	1.758 (1.703, 1.813)
total sugars	29.47 (28.78, 30.18)	35.79 (34.71, 36.9)	29.88 (29.18, 30.6)	28.75 (28.07, 29.44)	33.62 (32.61, 34.67)	28.8 (28.13, 29.5)

The box plots show that the contents of calcium (Figure 2B) and magnesium (Figure 2C) were not affected by nitrogen, but the content of magnesium was lower in bread wheat grain than in grains of spelt or emmer. The contents of iron and zinc were lowest in bread wheat grain (Figure 2D,E), while the content of iron was lower in emmer than in spelt grain. The contents of both minerals were also higher at 200 kg N/Ha than at 100 kg N/Ha. However, ANOVA showed significant differences between the contents of all minerals in the grain types and significant effects of nitrogen fertilisation on the contents of all minerals except magnesium, which showed a significant interaction due to a difference between nitrogen fertilisation in emmer only (Table 2). There were also significant differences between the contents of all minerals (except zinc in spelt and iron in all grain types) between the cultivars within each type, but no differences in the effects of nitrogen between the cultivars within the types (except for iron in spelt) (Table 2 and Appendix A). 

### 3.3. Total Phenolics

Phenolics are the most abundant phytochemicals in wheat grain [11]. The contents of total phenolics varied widely (Figure 2F), with significant differences between cereal types and cultivars within types (Table 2). There was an interaction between cereal type and nitrogen, with total phenolics being lower at 100 kg N/Ha for all cereal types apart from spelt where total phenolics were higher at 100 kg N/Ha than at 200 kg N/Ha.

### 3.4. Polar Metabolites

The contents of polar metabolites in the samples were determined by ^1^H NMR spectroscopy. This allowed the quantification of monosaccharide (glucose, fructose, galactose), disaccharide (maltose, sucrose) and trisaccharide (raffinose) sugars, organic acids (malic, acetic, fumaric), the sugar alcohols inositol and galactinol, the “methyl donors” choline and betaine and thirteen amino acids (alanine, aspartic acid, asparagine, glycine, glutamic acid, glutamine, γ-amino butyric acid (GABA), isoleucine, leucine, phenylalanine, tyrosine, tryptophan and valine). Data for all components are given in Appendix A, while selected components and groups of components are shown in Table 1 and Table 2 and Figure 3.

The contents of all metabolites and groups of metabolites overlapped between the cereal types, but differences between the ranges in the types are observed (Figure 3). Asparagine is of particular interest because it is the limiting factor for the formation of acrylamide during processing [12,13]. The contents of total amino acids (Figure 3A) and of asparagine (Figure 3B) were significantly lower in bread wheat and higher in spelt, with a significant effect of nitrogen. However, there were no differences in the effects of nitrogen between and within the three cereal types. 

The contents of total sugars (mono-, di- and trisaccharides) and total organic acids varied widely (Figure 3C,D), but both were significantly higher in emmer. Total organic acids were also significantly lower in bread wheat (Table 2). The contents of sugars were significantly affected by nitrogen, with no differences between the effects of nitrogen on the different cereal types or cultivars within the types (Table 2). The concentration of glycine betaine was about 10-fold greater than that of choline, which is typical for wheat [14]. Although the ranges overlapped (Figure 3E,F), they were significantly higher in spelt and lower in bread wheat, with little effect of nitrogen or variation within the types (Table 2).

Raffinose (the trisaccharide galactose, glucose, fructose) is a non-digestible and fermentable carbohydrate (being part of the FODMAP (fermentable oligosaccharides, disaccharides, monosaccharides and polyols) fraction) while galactinol (1-alpha-D-Galactosyl-myo-inositol) and inositol ((1*R*,2*S*,3*r*,4*R*,5*S*,6*s*)-Cyclohexane-1,2,3,4,5,6-hexol) are precursors in raffinose synthesis [15]. Raffinose (Figure 3I) accounted for about a quarter of the total sugars (Figure 3C) and was significantly lower in bread wheat than in emmer or spelt. The contents of inositol were about half of those of raffinose and were higher in spelt (Figure 3G). Galactinol was present at much lower concentrations and was significantly higher in emmer and lower in bread wheat than in spelt (Figure 3H). The contents of raffinose and inositol were affected by nitrogen level and, with the exception of galactinol in spelt, varied significantly between cultivars of the three cereal types (Table 2).

PCA analysis of the metabolite dataset showed partial separation of the three cereal types, based on 48.9% of the total variance (Figure 4A). To improve the discrimination between cereal types, we repeated the analysis using supervised multivariate analysis (orthogonal partial least squares discrimination analysis, OPLS-DA), selecting for differences between the cereal types (Figure 4C). This gave clear separation between the three types with the loadings plot (Figure 4F) showing that emmer was characterised by high contents of maltose, galactinol and glucose and spelt by higher levels of amino acids (glycine, asparagine, leucine, isoleucine and valine) compared with bread wheat. OPLS-DA was also used to separate the samples based on nitrogen level (Figure 4B), the loadings plot (Figure 4E) showing higher contents of amino acids at 200 kg N/Ha. These differences are also illustrated by the difference plots in Appendix A.

**Table 2 foods-12-00843-t002:** *p* values from ANOVA of minerals and selected metabolites in the three cereal types and cultivars.

	NLevel	Grain	Nlevel. Grain	Grain. Cultivar. Bread Wheat	Grain. Cultivar. Emmer	Grain. Cultivar. Spelt	Nlevel. Grain. Cultivar Bread Wheat	Nlevel. Grain. Cultivar Emmer	Nlevel. Grain. Cultivar Spelt

%N	**<0.001**	**<0.001**	0.475	**<0.001**	0.098	**<0.001**	0.704	0.212	0.642
log(Ca)	**<0.001**	**<0.001**	0.639	**<0.001**	**<0.001**	**<0.001**	0.672	0.195	0.495
log(Fe)	**<0.001**	**<0.001**	0.815	0.566	0.957	0.064	0.375	0.936	**0.023**
Mg	0.46	**<0.001**	**0.016**	**<0.001**	**0.03**	**<0.001**	0.228	0.351	0.749
Zn	**<0.001**	**<0.001**	**0.008**	**0.012**	**<0.001**	0.556	0.409	0.834	0.994

Total phenolics	0.407	**<0.001**	**<0.001**	**0.002**	**<0.001**	**<0.001**	0.253	0.516	0.153

raffinose	**0.005**	**<0.001**	**<0.001**	**<0.001**	**<0.001**	**0.001**	0.988	0.271	0.709
Log(asparagine)	**<0.001**	**<0.001**	0.352	**<0.001**	**0.022**	**<0.001**	0.624	0.34	0.78
Sqrt (glycine betaine)	0.358	**<0.001**	0.087	**<0.001**	**<0.001**	**<0.001**	**0.014**	0.951	0.305
choline	**<0.001**	**<0.001**	0.535	**<0.001**	**0.005**	**<0.001**	0.852	0.109	1
Sqrt (galactinol)	0.399	**<0.001**	0.061	**<0.001**	**<0.001**	0.605	0.935	0.146	0.951
inositol	**0.002**	**<0.001**	**0.009**	**<0.001**	**<0.001**	**<0.001**	0.883	0.06	0.678

log_e_(total amino acids)	**0.002**	**<0.001**	0.598	0.099	**<0.001**	**0.001**	0.095	0.672	0.464
total organic acids	0.163	**<0.001**	0.338	**<0.001**	**<0.001**	**<0.001**	0.159	0.771	0.551
log_e_ (total methyl donors)	0.529	**<0.001**	0.141	**<0.001**	**<0.001**	**<0.001**	**0.011**	0.905	0.395
log_e_(total sugars)	**0.005**	**<0.001**	0.397	**<0.001**	**<0.001**	**<0.001**	0.726	0.23	0.726

Statistically significant values (*p* < 0.05) are given in bold. Some variables required transformation, square root (Sqrt) or log_e,_ to meet the assumptions of the analysis.

**Figure 3 foods-12-00843-f003:**
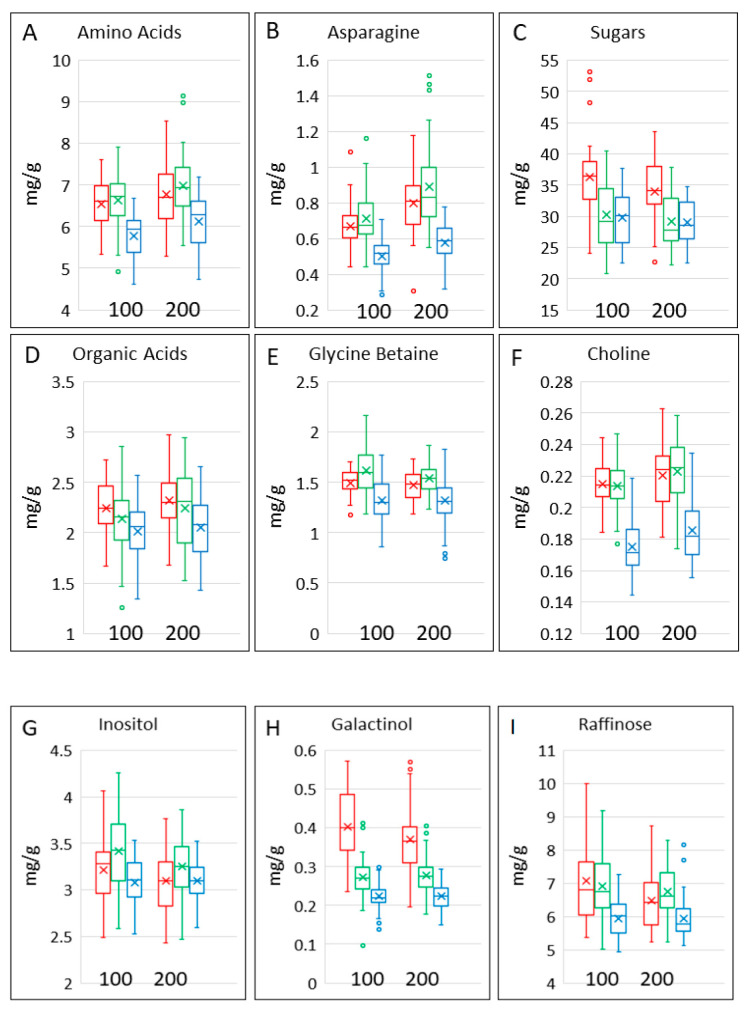
Contents of selected polar metabolites and groups of metabolites in grains of the three cereal types grown in four environments. Colour code: red, emmer; green, spelt; blue, bread wheat. The bar shows the range of the whole data set. The box shows the middle two quartiles, separated by the horizontal line, which is the median, and the vertical lines are the upper and lower quartiles, respectively. Outliers are shown as circles. The x is the mean average. All analyses are expressed on a dry weight basis.

**Figure 4 foods-12-00843-f004:**
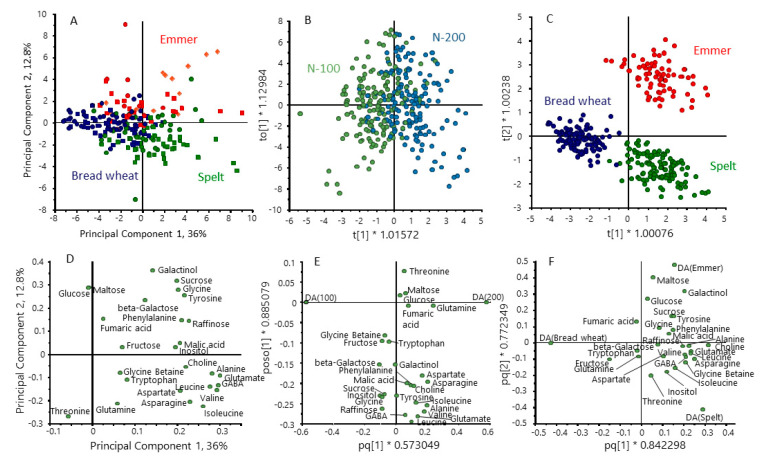
Multivariate analysis of the contents of polar metabolites in grains of the three cereal types grown in four environments. Principal component analysis PCA (**A**) and orthogonal partial least squares discrimination analysis (OPLS-DA) (**B**,**C**) selecting for differences between nitrogen treatments (**B**) and cereal types (**C**). (**D**–**F**) loading plots for (**A**–**C**), respectively. Colour code: red, emmer; green, spelt; blue, bread wheat.

### 3.5. Dietary Fibre

The contents of AX and β-glucan in the grain types are shown in Figure 5A,B and in Table 3. The contents of both AX and β-glucan were lower in emmer grain, with AX being highest in bread wheat grain and β-glucan highest in spelt grain. Hence, the ratio of AX: β-glucan was lower in spelt grain than in the other cereals (Figure 5D). The combined contents of these two components were also highest in bread wheat grain (Figure 5C), reflecting the fact that the content of AX was about three- to four-fold greater than that of β-glucan. There was significant variation in the contents of AXOS between the cultivars of the three types of wheat and of β-glucan between the cultivars of bread wheat and emmer, but no significant effects of nitrogen fertilisation (Table 4).

AX and β-glucan were determined by enzyme fingerprinting, which uses enzymes (endoxylanase and lichenase, respectively) to digest the polymers to release oligosaccharides separated and quantified by HP-AEC. The oligosaccharides released have defined structures, and their proportions, therefore, provide information on the structures of the polymers. In the case of AX, the oligosaccharides (AXOS) comprise chains of one to five xylose residues, one or more of which may be substituted with either one or two arabinose residues. The ratio of monosubstituted to disubstituted AX may affect the properties of the molecules and is generally higher in spelt and bread wheat than in emmer (Figure 6F). β-glucan comprises linear chains of glucose molecules linked predominantly by β(1-4) bonds. However, these β(1-4) bonds are interspersed with β(1-3) bonds that generally occur every three to four glucose residues, although some longer stretches of β(1-4) linked glucose residues (up to 14) have been reported. The distribution of β(1-3) bonds results in conformational changes in the linear glucan molecules, which affect their solubility and viscosity. Lichenase is a type of β-glucanase that releases mainly gluco-oligosaccharides (GOS) of three or four glucose residues (called G3 and G4), reflecting the relative abundances of β(1-3) and β(1-4) bonds. The ratio of G3:G4 GOS is higher in emmer and lower in spelt than in bread wheat (Figure 5E, Table 3).

**Table 3 foods-12-00843-t003:** Means and 95% confidence intervals (in parentheses) of contents and compositions of arabinoxylan and β-glucan in grain of the three types of wheat grown with 100 and 200 kg N/Ha.

	100 kg N/Ha			200 kg N/Ha		
	Bread	Emmer	Spelt	Bread	Emmer	Spelt
TOT-AX	29.06 (28.18, 29.96)	20.17 (19.43, 20.93)	27.33 (26.51, 28.18)	29.61 (28.72, 30.53)	19.88 (19.16, 20.64)	27.26 (26.44, 28.11)
TOT-BG	8.146 (7.869, 8.432)	5.617 (5.386, 5.859)	9.256 (8.942, 9.581)	7.63 (7.371, 7.898)	5.142 (4.93, 5.363)	8.521 (8.232, 8.82)
ratio G3:G4 GOS	2.429 (2.395, 2.463)	2.54 (2.497, 2.583)	2.268 (2.234, 2.302)	2.381 (2.347, 2.415)	2.477 (2.434, 2.52)	2.252 (2.218, 2.286)
ratio TOT-AXOS: TOT-BG	3.596 (3.516, 3.677)	3.619 (3.516, 3.723)	2.968 (2.887, 3.048)	3.915 (3.834, 3.996)	3.895 (3.792, 3.999)	3.212 (3.131, 3.293)
ratio M:D AXOS	2.061 (2.035, 2.088)	1.865 (1.834, 1.897)	2.095 (2.068, 2.122)	2.032 (2.005, 2.058)	1.85 (1.818, 1.881)	2.071 (2.044, 2.097)

**Figure 5 foods-12-00843-f005:**
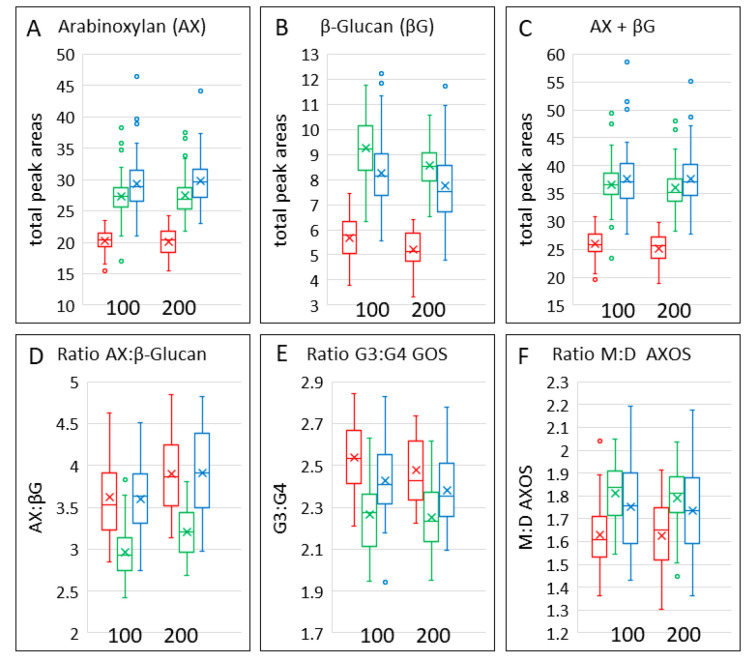
Contents, ratios and structures of arabinoxylan and β-glucan in grains of the three cereal types grown in four environments. Colour code: red, emmer; green, spelt; blue, bread wheat. The box shows the middle two quartiles, separated by the horizontal line, which is the median, and the vertical lines are the upper and lower quartiles, respectively. Outliers are shown as circles. The x is the mean average.

Differences in the structures of AX and β-glucan in the cereal types are illustrated by the multivariate analysis in Figure 6. OPLS-DA confirmed that there were no effects of nitrogen on AX and β-glucan structure (Figure 6B) but gave clear separation of the cereal types (Figure 6C), with the loadings plot (Figure 6F) showing that spelt differed from emmer and bread wheat in having higher proportions of G3 and G4 GOS and bread wheat higher proportions of substituted AXOS These differences are also illustrated by the difference plots in Appendix A

**Table 4 foods-12-00843-t004:** *p*-values from ANOVA of the proportions of AXOS and GOS in the three cereal types and cultivars.

	Nlevel	Grain	Nlevel. Grain	Grain. Cultivar. Bread wheat	Grain. Cultivar. Emmer	Grain. Cultivar. Spelt	Nlevel. Grain. Cultivar. Bread wheat	Nlevel. Grain. Cultivar. Emmer	Nlevel. Grain. Cultivar. Spelt
log_e_ (TOT-AXOS)	0.855	**<0.001**	0.514	**<0.001**	**<0.001**	**<0.001**	0.42	0.848	0.926
log_e_ (TOT-BG)	**0.002**	**<0.001**	0.758	**<0.001**	**<0.001**	0.085	0.659	0.501	0.76
ratio G3:G4 GOS	**0.032**	**<0.001**	0.427	0.509	0.197	**<0.001**	0.711	0.513	0.423
ratio TOT -AXOS: TOT-BG	**<0.001**	**<0.001**	0.655	**<0.001**	0.121	**<0.001**	0.715	0.218	0.564
Sqrt (ratio M:D AXOS)	0.093	**<0.001**	0.899	**<0.001**	**<0.001**	**<0.001**	0.249	0.775	0.902

Statistically significant values (*p* < 0.05) are given in bold. Some variables required transformation, square root (Sqrt) or log_e_, to meet the assumptions of the analysis.

**Figure 6 foods-12-00843-f006:**
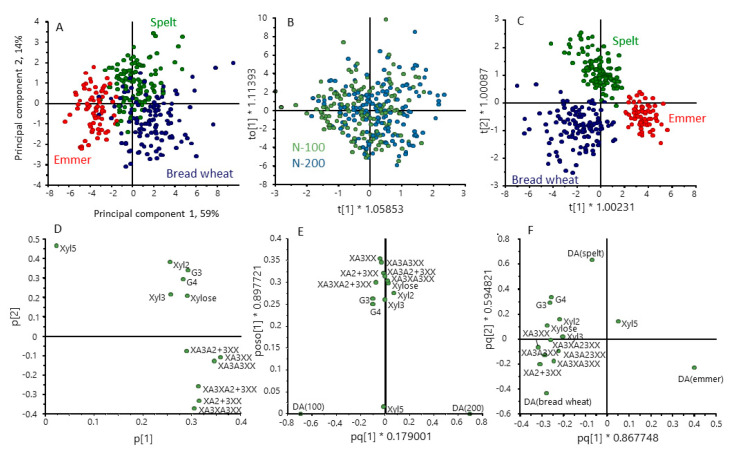
Multivariate analysis of the contents of AXOS and GOS in grains of the three cereal types grown in four environments. Principal component analysis PCA (**A**) and orthogonal partial least squares discrimination analysis (OPLS-DA) (**B**,**C**) selecting for differences between nitrogen treatments (**B**) and cereal types (**C**). (**D**–**F**) loading plots for (**A**–**C**), respectively. Colour code: red, emmer; green, spelt; blue, bread wheat.

## 4. Discussion

We have carried out comparative analyses of the grain of five cultivars each of bread wheat and spelt and three cultivars of emmer, focusing on components that may contribute to health effects. The cultivars selected were all commercially available in Germany at the time of the study, and analyses of flours blended from commercial grain samples and of doughs and breads produced from the blended flours using sourdough and yeast-based systems have been reported elsewhere [16]. The cultivars were grown in replicated randomised field trials at sites in the UK and Netherlands for two years, giving four environments (sites and years). Furthermore, two levels of nitrogen fertiliser were applied to reflect the commercial use of low and high input systems for ancient wheats (emmer, spelt) and modern bread wheat, respectively.

Because only small numbers of cultivars were compared, the results cannot be taken to represent the full range of diversity within the three types of wheat. Nevertheless, wide variation within each type was observed, resulting from the effects of genotype, environment and nitrogen fertilisation. However, because only two sites and years were compared, it was not possible to calculate the separate contributions of genotype, environment and G x E interactions, and the year/site combinations were, therefore, treated as “environments”.

The variation in composition within emmer, spelt and bread wheat resulted in overlapping ranges in the contents of all components determined: protein, minerals, polar metabolites and AX and β-glucan fibre. Nevertheless, some statistically significant differences between the three cereal types were observed.

The lower protein content of modern bread wheats observed here is well-established and considered to result from “yield dilution”; the higher yields of modern wheats result from an increased accumulation of starch that dilutes other components [17]. Similarly, modern semi-dwarf bread wheats are known to have lower contents of Fe, Zn and Mg. This may result from the effects of the semi-dwarf phenotype, perhaps combined with some yield dilution [18,19]. The modern bread wheats were also significantly lower in asparagine, the precursor of acrylamide, which formed during processing [12] but also had lower contents of glycine betaine and choline, which have benefits for cardiovascular health by reducing the concentration of homocysteine in blood [20,21]. Hence, it is not possible to conclude that any of the three types of cereal is consistently “better” in terms of its content of polar metabolites.

Published values for the contents of total dietary fibre in whole grains of bread wheat range between 11.5–15.5% dry weight, of which about half (5.5–7.4% dry weight) is AX, with a lower content of β-glucan (0.51–0.96%) [22]. White flour contains significantly less fibre (due to the removal of the bran), about 4% dry weight, with AX and β-glucan accounting for about 70% and 20% of the total, respectively [23].

In the present study, bread wheat had higher contents of AX and β-glucan than emmer and higher AX content than spelt. Although spelt was higher in β-glucan, this component was present in lower concentrations than AX, and hence, the sum of the two types of fibre was highest in bread wheat. A meta-analysis of dietary fibre components in whole grain also showed slightly lower contents of AX in spelt than in bread wheat, with a wider range [24]. Some differences in fibre structure were also observed, with spelt having a lower ratio of G3:G4 GOS released from β-glucan while emmer had a lower ratio of monosubstituted:disubstituted AXOS released from AX. The significance of these differences for the behaviour of the AX and β-glucan fractions in foods and in the gastrointestinal tract is not known.

The bread wheats also had lower contents of raffinose, which is not absorbed in the human small intestine but rapidly fermented in the colon, forming part of the FODMAP fraction, which may contribute to discomfort due to gas production in individuals suffering from non-celiac wheat sensitivity and irritable bowel syndrome (IBS) [25]. However, the relevance of this to symptom control is limited as the major FODMAP fraction in wheat is fructans, which were not measured in the present work.

The effects of nitrogen fertilisation were also determined as it is usual to grown ancient and modern wheats under low input and intensive production systems, respectively. High nitrogen resulted in higher contents of minerals (iron, zinc and magnesium), as reported in a number of studies [26]. Similarly, a significant positive relationship between free asparagine content and total grain protein content has been reported [27,28]. Although only small effects of nitrogen on other components were observed, ANOVA showed that these varied between cereal types. ANOVA also showed interactions between N level and the proportions of AXOS and GOS released in all three types of wheat but no interactions with cultivars within the types.

A number of other studies have compared genotypes of modern and ancient wheats. For example, a series of studies compared the agronomic performance, yield, grain quality traits and contents of a range of “bioactive” components in 15 cultivars each of einkorn, emmer, spelt, bread wheat and durum wheat grown on four sites, with nitrogen fertilisation levels varying between wheat types to reflect commercial practice [29,30,31]. However, the relevance of such reported differences in composition to human health remains unclear.

This is, at least in part, due to the fact that the relevance of the parameters measured to human health has not been established. For example, although differences between the in vitro antioxidant capacity of cereal flours have been reported [32,33,34], these cannot be generalized to imply health benefits in humans in vivo [35]. Similarly, glucose released during the digestion of flour in vitro cannot be used to predict glycaemic response in vivo, which is determined by the competing effects of the appearance of glucose and the disappearance (cellular uptake) of glucose in blood [36,37,38]. As a result, although the glycaemic index (GI) calculated based on in vitro digestion may correlate with that determined in vivo, the absolute values may differ [39]. Such calculations based on in vitro digestion have been reported to result in over-estimation of the in vivo GI by 22% to 50% [40]. Differences in food structure resulting from processing will also affect oral mastication and gastrointestinal digestion. For example, the high density of pasta results in a significantly lower glycaemic response than those of flour and bread [41].

Finally, statistically significant differences in the compositions of cereal flours may not represent significant biological differences when considered in the context of processed food consumed as part of a mixed meal and with other foods consumed over a 24 h intake period.

Comparisons are also often reported using analytical data from different studies and/or data from published food composition tables (for example, [42,43]). This is clearly not valid as the cultivars, growth environments, agronomic practices and methods used for sample preparation and analysis will affect the results obtained.

Taking these factors into account, it is not surprising that different conclusions have been drawn on the health benefits of ancient compared with modern wheats. Thus, whereas Shewry and Hey [24] concluded that based on a comparison of grain samples grown and analysed under the same conditions, there is little evidence that ancient wheats are more “healthy” than modern wheats, Serban et al. [43] suggested that ancient wheat species have health benefits in relation to their nutraceutical composition.

In the context of the data presented here, the small differences in mean compositions of bread wheat, emmer or spelt and their overlapping quantitative ranges are unlikely to result in significant differences in health outcomes, with a possible exception being mineral micronutrients (zinc and iron, which are subject to low intakes in many population groups, with wheat being a significant dietary source [44].

## Figures and Tables

**Figure 1 foods-12-00843-f001:**
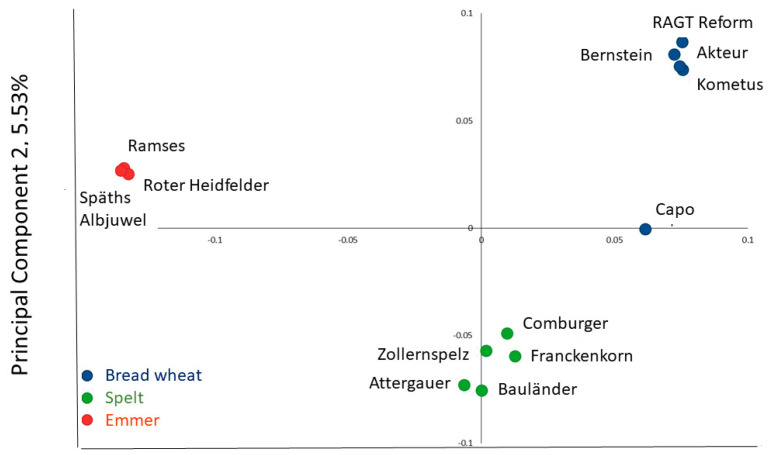
Genomic relationships of the 13 genotypes, illustrated by Principal Component Analysis of markers determined using the Axiom HD Genotyping Array (comprising 819,571 SNP markers).

## Data Availability

Full datasets are available from the corresponding author on request.

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
