# Peer review of "Comparative Compositions of Grain of Bread Wheat, Emmer and Spelt Grown with Different Levels of Nitrogen Fertilisation"

_foods, 2023, doi:10.3390/foods12040843_

Round 1
Reviewer 1 Report
I find this paper very well written and based on reliable data. The authors, very honestly, do not pretend to draw absolute conclusions about the comparison between “ancient” and “modern” wheats, since a small number of cultivars and species in two locations and two years were taken into consideration but, in my opinion, this is MUCH MORE than the usual number I am used to read.
In fact, in a context where I often read papers comparing “ancient” and “modern” wheats by using only one genotype/cultivar per group and one environment in one growing condition in a single experiment, this paper absolutely stands out and reveals that the difference must not be searched in the two groups but mostly in the single genotypes grown in specific conditions, despite the year of release.
Author Response
This review is very supportive and no response is required
Reviewer 2 Report
Line 61: cluding temperature and water availability, soil type and agronomy) and the interactions….I propose to write a farming system
Line 72-73: and higher contents of proteins which may lead to adverse reactions and disease... What diseases or reactions? Does the protein content increased or the amino acid profile changed?
Line 91: Please add years of grain harvest.
Line 98: …were milled in two stages… Please add detailed information on the preparation of grain for milling (conditioning, moistening before milling), the method and conditions of milling (name of the mill, producer).
Line 115: I recommend citing the source of literature recommended for preparing of the manuscript.
Line 155: In what range of concentrations should the ferulic acid standard curve be prepared?
Figure 2: Is the protein content (%) expressed on a grain dry matter?
Line 208-211: This should be the description under the Figure 2.
Line 216-224: The results discussed are for wheat grains and other species tested, not for wheat and other species (whole plants).
Line 251: Has there been a change in the content of limiting amino acids under the influence of the factors used?
Line 301-308: The results discussed are for wheat grains and other species tested, not for wheat and other species (whole plants).
Line 334-336: This should be the description under the Figure 5.
Author Response
Line 61: including temperature and water availability, soil type and agronomy) and the interactions….I propose to write a farming system
Done
Line 72-73: and higher contents of proteins which may lead to adverse reactions and disease... What diseases or reactions?
Done
Does the protein content increased or the amino acid profile changed?
Amino acid composition was not measured
Line 91: Please add years of grain harvest.
Done
Line 98: …were milled in two stages… Please add detailed information on the preparation of grain for milling (conditioning, moistening before milling), the method and conditions of milling (name of the mill, producer).
Whole grain samples at about 14% water content were milled in two stages: firstly, a Retsch ZM 200 Model Ultra-Centrifugal Mill using a 0.5 mm ring sieve and then a Glen Creston Ball Mill using 5 ball bearings in a 5cm diameter cannister for 4 minutes for each sample.
Line 115: I recommend citing the source of literature recommended for preparing of the manuscript.
Done: lovegrove et al (2013) added.
Line 155: In what range of concentrations should the ferulic acid standard curve be prepared?
The absorbance at 725nm was then measured (Jenway 6715 UV/Vis spectrophtometer) and the concentration of phenolics calculated using ferulic acid (Sigma-Aldrich) and a standard curve from 20, 40, 100, 150 and 200 µg/ml with three technical reps of each point.
Figure 2: Is the protein content (%) expressed on a grain dry matter?
Dry wt: added to Figure legend
Line 208-211: This should be the description under the Figure 2.
This has been closed up
Line 216-224: The results discussed are for wheat grains and other species tested, not for wheat and other species (whole plants).
Line 251: Has there been a change in the content of limiting amino acids under the influence of the factors used?
This was not determined
Line 301-308: The results discussed are for wheat grains and other species tested, not for wheat and other species (whole plants).
This is corrected
Line 334-336: This should be the description under the Figure 5.
This has been closed up
Reviewer 3 Report
The authors evaluated the effect of nitrogen application on the proximate composition of three types of wheat, i.e., bread wheat, spelt, and emmer wheat. The need to evaluate the proximate composition after N fertiliser application is very important to understand the role of wheat crop improvement/breeding programmes on the nutritional value of wheat. It will aid in the development of more weather-resistant, nutritionally dense wheat varieties.
A simple table with data collected for control, 100 and 200 kg of N fertiliser for proximate composition, protein and sugar composition, and so on, with statistical differences (mean, average, SEs), would be a better choice.
A simple two-way ANOVA with F values and p values (* = p≤ 0.05 0r *= p≤ 0.005) to answer the effect of cultivars, environment and nitrogen dose application on various parameters will be easier to understand.
Cultivars; Nitrogen dose; Environment; Cult X N; Cult x Env; Cultivar x Nitrogen x Environment.
Protein; carbohydrates (both starch and NSP/pentosans); organic acids; amino acids, etc., nitrogen would be more impressive. Box plots look good, but they require a very critical understanding of statistics. So data in simple tables would be more useful. Tables 1, 2, and 3 data can be moved to supporting files.Original data presented in box plots can be presented in tables with means and SEMs, which will enhance the readability of MS.
This type of presentation looks good in articles published Nature, Science, and Cell Type Research Journals, but presenting the data in a more simple way herein may be more impressive,.
PCA data is good and may be included in the main MS.
Author Response
We appreciate the reviewer’s detailed comments on the data presentation.
The paper is based on large and complex datasets which are analysed by a professional statistician (Jessica Hood, who is an author) and the use a simpler treatment of the data (such as 2-way ANOVA) is not statistically valid.
However, we will replace Tables 1 and 3 with simpler Tables of back-transformed data which are more readily interpreted by readers who are not experts in statistics.
We will retain the Figures because they display a lot of information (on means, ranges, outliers etc) in a clear format.
Reviewer 4 Report
Comparative compositions of grain of bread wheat, emmer and spelt grown with different levels of nitrogen fertilization
Comments:
Improve abstract. Summarized whole article in abstract and add importance of this research.
Key words could be better and preferable to have an alphabetical sequence.
Add conclusive lines in the end of abstract that should summarize the whole study.
Add importance and rationale of your study in the end of introduction.
Explain ‘how nitrogen fertilization effects the composition of grains’.
Recheck font size of line no 13.
Please give the composition before starting the methodology.
Explain the levels of nitrogen fertilization along with its role in the growth.
Add rational and study limitations.
Add reference for mineral analysis in material and methods.
Add units of minerals and protein in table no 1.
Add the calculated amount of ferulic acid and dietary fibers present in grains.
Draw table for metabolites present in grains.
Explain all the methods in details also mention the model and details of NMR.
Statistical analysis is very confusing please mention the study design and test that has been applied in this study.
Add conclusion separately at the end of the article and add major findings to conclude the whole article.
Recheck the line spacing and font size from line no 454 to 502.
Give results of your NMR in figures.
Repetition of some words in whole article please avoid it.
Please format all references and internal citations according to journal style.
Read whole article and check its grammar and language.
Author Response
This review requested extensive revisions. However, most of the requests were already satisfied in the submitted paper.
Add conclusive lines in the end of abstract that should summarize the whole study: the abstract includes a conclusion (lines 29-31)
Key words could be better and preferable to have an alphabetical sequence. We think the words selected are appropriate but will arrange them alphabetically
Add importance and rationale of your study in the end of introduction.: this was already described in detail in the last paragraph of the introduction (lines 75-85)
Explain ‘how nitrogen fertilization effects the composition of grains’: this is not possible in a research paper (as opposed to a review article).
Recheck font size of line no 13: done
Please give the composition before starting the methodology: I do not know what is being requested here, beyond the results reported?
Explain the levels of nitrogen fertilization along with its role in the growth: the rational for the selected levels of nitrogen was clearly explained (lines 78-82). It is not possible to describe the role of N in plants in a research article.
Add rational and study limitations: these were clearly discussed (lines 355-370)
Add reference for mineral analysis in material and methods: this was included (lines 137-145)
Add units of minerals and protein in table no 1: this will be done.
Add the calculated amount of ferulic acid and dietary fibers present in grains: these were given in tables and figures, except that total phenolics were determined as ferulic acid equivalents (not ferulic acid).
Draw table for metabolites present in grains: full datasets were already provided in Table and Supplementary Tables 1-3
Explain all the methods in details also mention the model and details of NMR: these were already included in the (lines 119-135)
Statistical analysis is very confusing please mention the study design and test that has been applied in this study: the study design and the statistics are clearly described (lines 156-171)
Add conclusion separately at the end of the article and add major findings to conclude the whole article: this was included (lines 449-453)
Recheck the line spacing and font size from line no 454 to 502: this has been done
Give results of your NMR in figures: selected NMR results are shown in Figure 2 with full results in Table 1 and Supplementary Tables S1, S2 and S3
Repetition of some words in whole article please avoid it: we aim to use the correct words where they are required.
Please format all references and internal citations according to journal style.: this was already done
Read whole article and check its grammar and language: I do not consider that this is required and it was not mentioned by other reviewers. I have written over 500 papers and a book in English which is my native language.
Round 2
Reviewer 4 Report
Quality of article is not suitable.
Still i think there are a lot of problem that should be descuss.